

# A quasi-randomised, controlled, feasibility trial of GLITtER (Green Light Imaging Interpretation to Enhance Recovery)—a psychoeducational intervention for adults with low back pain attending secondary care

Emma L. Karran[1,2], Susan L. Hillier[1], Yun-Hom Yau[2], James H. McAuley[3,4] and G. Lorimer Moseley[1,3]

[1] School of Health Sciences, University of South Australia, Adelaide, South Australia, Australia
[2] Royal Adelaide Hospital, Adelaide, South Australia, Australia
[3] Neuroscience Research Australia, Sydney, New South Wales, Australia
[4] School of Medical Sciences, Faculty of Medicine, University of New South Wales, Sydney, New South Wales, Australia

Corresponding authors
Emma L. Karran,
emma.karran@mymail.unisa.edu.au
G. Lorimer Moseley,
Lorimer.Moseley@gmail.com,
lorimer.moseley@unisa.edu.au

## ABSTRACT

**Background**. Although it is broadly accepted that clinicians should endeavour to reassure patients with low back pain, to do so can present a significant clinical challenge. Guidance for how to provide effective reassurance is scarce and there may be a need to counter patient concerns arising from misinterpretation of spinal imaging findings. 'GLITtER' (Green Light Imaging Intervention to Enhance Recovery) was developed as a standardised method of communicating imaging findings in a manner that is reassuring and promotes engagement in an active recovery. This feasibility study is an important step towards definitive testing of its effect.

**Methods**. This feasibility study was a prospective, quasi-randomised, parallel trial with longitudinal follow-up, involving sampling of patients attending a spinal outpatient clinic at a metropolitan hospital. English speaking adults (18–75 years) presenting to the clinic with low back pain and prior spinal imaging were considered for inclusion. Eligible patients were allocated to receive a GLITtER consultation or a standard consultation (as determined by appointment scheduling and clinician availability), and were blinded to their allocation. Full details of the GLITtER intervention are described in accordance with the *Tidier* template. Follow-up data were collected after 1 and 3 months. The primary outcome of this study was the fulfillment of specific feasibility criteria which were established *a priori*. Determination of a sample size for a definitive randomised controlled trial was a secondary objective.

**Results**. Two hundred seventy-six patients underwent preliminary screening and 31 patients met the final eligibility criteria for study inclusion. Seventeen participants were allocated to the intervention group and 14 were allocated to the control group. Three month follow-up data were available from 42% of the 31 enrolled participants ($N = 13$, six intervention, seven control). Feasibility indicators for consent, resource burden and acceptability of the GLITtER intervention were met, however participant recruitment was slower than anticipated and an acceptable follow-up rate was not achieved.

**PeerJ** ________________________________________

**Conclusions**. Failure to achieve pre-specified recruitment and follow-up rates were important outcomes of this feasibility study. We attribute failure to issues that are likely to be relevant for other clinical trials with this population. It is realistic to consider that these challenges can be overcome through careful strategy, ample funding and continued partnership with health care providers.

**Trial registration**. The trial was registered on the Australian and New Zealand Clinical Trials Registry on 28/2/2017 (ACTRN12617000317392).

## INTRODUCTION

Patients with low back pain (LBP) attending secondary care settings can be characterised by high pain intensity, poor function and higher frequency of poor prognosis than patients attending primary care (*Karran et al., 2017b*; *Morsø, et al., 2013*; *Morsø, et al., 2014*). Consequently, their contribution to the massive burden of LBP (*Vos et al., 2016*) is likely to be significant. General practitioners (GPs) frequently refer their patients to specialist secondary care clinics for surgical opinion (*ABS, 2015*; *Coulter, 1998*)—however surgery is recommended for only a small proportion of cases (*Li & Yen, 2010*; *Robarts et al., 2017*). For the remainder, clinicians are generally challenged to provide time-efficient, guideline-based, conservative care recommendations. Screening tools have been used to determine risk and optimal pathways for back pain patients in primary care, but systematic review data (*Karran et al., 2017a*) are underwhelming and they seem to offer little benefit within the context of secondary care (*Karran et al., 2017b*).

There is widespread current agreement in LBP clinical practice guidelines that clinicians should deliver high quality information as a key component of their management (*O'Connell et al., 2016*; *Wong et al., 2017*). In particular, patients should be reassured that their condition is not likely to be serious, that a favourable outcome is usual and that activity levels should be resumed as soon as possible (*NICE, 2016*; *Koes et al., 2010*; *Qaseem et al., 2017*; *Stochkendahl et al., 2017*). While patient reassurance can be considered a central tenet of this approach, practical guidance for how to effectively deliver reassurance in the clinical setting is lacking (*Hasenbring & Pincus, 2015*; *Traeger et al., 2015*). Furthermore, the impact of providing reassuring information on pragmatic outcomes appears to be poorly understood (*Pincus & McCracken, 2013*; *Traeger et al., 2017*).

Counter to clinicians' efforts to reassure, the communication of spinal imaging findings have been suggested to have the potential to increase patients' fear of re-injury and reduce their likelihood of a good outcome (*Roland & Van Tulder, 1998*). Adverse effects of early imaging of the lumbar spine have also been reported (*Graves et al., 2012*; *Webster & Cifuentes, 2010*). That most LBP patients attending secondary care present with spinal images (*Sears et al., 2016*) and that these images are routinely considered during the

consultation, provides opportunity for careful consideration of how imaging results are interpreted and communicated in this setting.

Recent evidence reveals a high prevalence of common degenerative features in the imaging reports of asymptomatic adults (*Brinjikji et al., 2014*). This indicates that many of these features—particularly when found incidentally—should not be considered pathological and instead be regarded as normal age-related changes. Coherent with this interpretation is that degenerative findings have been found to be poorly associated with current pain (*Brinjikji et al., 2014*; *Carragee et al., 2006*; *Suri et al., 2014*) and prognosis (*Carragee et al., 2005*; *De Schepper et al., 2016*; *Jensen et al., 2014*; *Steffens et al., 2014*). Considering this evidence, we developed and tested an intervention framework based on using imaging findings for clinical benefit (*Karran et al., 2018*). This psycho-educational intervention, referred to as 'GLITtER' (Green Light Imaging Intervention to Enhance Recovery), involves a standardised method of communicating radiological findings in a manner designed to reassure patients and promote engagement in an active recovery.

A feasibility trial, conducted in the spinal outpatient setting at an Australian metropolitan hospital, is the crucial first step towards definitive testing of this intervention. The primary aim of this study was to determine the feasibility of recruitment and retention, assessment procedures, implementation and acceptability of the GLITtER intervention for LBP patients attending a spinal outpatient clinic. Secondary objectives included the identification of modifications needed in the design of a larger effectiveness trial and provision of data to enable calculation of an appropriately powered sample for a subsequent effectiveness trial.

## METHODS

### Trial approval, registration and reporting

The Research Ethics Committees at the Royal Adelaide Hospital (protocol no. 150308) and the University of South Australia (protocol no. 0000034887) provided approval for this study. The full protocol was pre-registered on the Open Science Framework (https://osf.io/8zrq3/) and the trial was registered on the Australian and New Zealand Clinical Trials Registry (http://www.anzctr.org.au, registration number: ACTRN12617000317392). Trial registration included details of all items from the World Health Organisation Trial Registration Data Set (see Supplemental File 1). This study has been reported according to the **CONSORT 2010 guideline** for transparent and quality reporting of randomised pilot and feasibility trials (http://www.consort-statement.org) (see Supplemental File 2).

### Study design

This investigation was a prospective, quasi-randomised feasibility trial with longitudinal follow-up, involving sampling of patients attending the Spinal Assessment Clinic (SAC). The study also adopted an adaptive trial design, whereby modifications could be made during its conduct with the purpose of increasing the probability of success of the study procedure or the intervention. Adaptations were made during recruitment of the first one-third of participants, as pre-specified in the protocol (https://osf.io/8zrq3/).

## Study setting and participants

The study was conducted in the SAC, which operates in the Spinal Outpatient Department at a large metropolitan hospital in South Australia. The SAC is a Physiotherapist-led clinic attended by patients who warrant non-urgent consultation, as identified by a paper-based triage procedure prior to appointment scheduling. All patients aged between 18 and 75 years who were scheduled to attend the SAC with a lumbar spine disorder were considered for inclusion. To be eligible, patients were required to be able to speak and understand (verbal and written) English and have access to recent images of their lumbar spine. Baseline pain duration was not an eligibility consideration, however it was anticipated that patients presenting to the SAC were likely to have experienced pain for longer than 3 months, based on usual minimum timeframes for appointment scheduling. Patients with a history of lumbar spine surgery were excluded.

## Intervention

Participants allocated to the intervention group received a 'GLITtER' consultation, integrated into the standard SAC consultation. Participants allocated to the control group received a 'standard consultation'. Full details of the consultations are provided in Table 1 in accordance with the *TIDieR* **template** for intervention description and replication (see http://www.equator-network.org). In brief—a routine, or 'standard' consultation involves comprehensive patient assessment, review and interpretation of relevant investigations and discussion of management recommendations. A key objective of the patient-centered interaction is identification of potential surgical candidates, or the guidance of patients appropriate for continued conservative care towards community-based options. The GLITtER consultation includes all components of the standard consultation but is enhanced by the implementation of a standardised framework through which imaging findings are interpreted (designed to optimise patient reassurance), includes provision of take-home resource, emphasizes the need for an *active* recovery, and offers links to further information.

## Staff training

Two SAC clinicians (clinicians 1 and 2) volunteered to deliver the GLITtER intervention and were provided with training. Two clinicians (clinicians 3 and 4) remained naïve to GLITtER and agreed to deliver the control intervention. A 3-stage training process was conducted for clinicians 1 and 2. The initial stage involved a meeting to provide background, rationale and an overview of the planned intervention. Within this session, a neurosurgeon (YHY) led instruction and discussion of the imaging interpretation strategy developed for GLITtER. The second stage involved providing clinicians 1 and 2 with the framework for the GLITtER intervention, with instruction to undertake self-directed familiarization with the content (see Supplemental File 3). The third stage was a face-to-face session with the principal researcher to discuss the intervention framework and practice strategies for clinical implementation. The total time for training and familiarisation was 2–2.5 h.

Karran et al. (2018), *PeerJ*, DOI 10.7717/peerj.4301

**Table 1** Description of the intervention and control consultations (consistent with the TIDieR reporting checklist (see www.equator-network.org)).

| | GLITtER consultation | Standard consultation |
|---|---|---|
| WHY | **Goal:** To supplement and enhance the standard SAC consultation with a strategy designed to optimise patient reassurance and facilitate engagement in an active recovery. | **Goal:** To provide comprehensive assessment and management of patients with LBP attending the SAC. Key objectives are to identify potential surgical candidates and to transfer the care of non-surgical candidates to General Practitioner supported community based care *Bear, Orlando & Kumar (2016)*. |
| WHAT | **Materials:**<br>• GLITtER Framework used for clinicians training (see Supplemental File 3)<br>• Visual aid A (used during intervention delivery): graph of prevalence of degenerative features in asymptomatic adults (see Supplemental File 7)<br>• Take-home information resource (see Supplemental File 8). This was designed as a series of 4 posters to be displayed one week at a time.<br>• Links to online information (incorporated into take-home resource and delivered via smartphone text messages):<br>  ○ The truth about back pain (https://www.youtube.com/watch?v=b-cBtPSf0Hc)<br>  ○ Tame the Beast (https://www.youtube.com/watch?v=ikUzvSph7Z4)<br>  ○ How to start exercising and stick to it (https://www.helpguide.org/articles/healthy-living/how-to-start-exercising-and-stick-to-it.htm)<br>• Understanding pain in less than 5 min (https://www.youtube.com/embed/qEWc2XtaNwg)<br>• Letter to General Practitioner (see Supplemental File 9) | **Materials:**<br>• No physical or informational materials are routinely used or provided |
| | **Procedures:**<br>• All procedures implemented in the standard consultation were included in the GLITtER consultation.<br>• Additional procedures (unique to the GLITtER consultation):<br>  i. Provide detailed information about 'normal', age-relevant imaging findings and involve visual aid A.<br>  ii. In addition to explaining patient's imaging findings, explain that: Scans (on their own) do not explain much about:<br>    - Your current pain (e.g., why you have good days and bad).<br>    - The activity you are capable of, or<br>    - How likely you are to recover (because the changes on your scans will still be there when your pain goes away)<br>  iii. Re-interpret imaging findings, highlighting 'positive' features. E.g.<br>    - Demonstrate spinal features that offer structural stability and emphasise the inherent strength of the spine.<br>    - Demonstrate musculature and joints–structures that need movement to be optimally healthy.<br>  iv. Promote using the 'TICK list' as a strategy for increasing planned activity/exercise (see Supplemental File 8).<br>  v. Introduce patient to take-home information (see Supplemental File 8).<br>  vi. Request patient completion of GLITtER checklist (See Supplemental File 10) and discuss further if required.<br>  vii. Text message follow-up: 4× (brief) weekly SMS messages prompted participants to display/read the relevant poster, and provided an active link to the online information recommended on the poster. | **Procedures:** A standard consultation involves:<br>• Subjective examination/patient history.<br>• Physical examination, including neurological assessment.<br>• Review of imaging and relevant investigations.<br>• Discussion of relevant findings (from examination and investigations).<br>• Discussion of management recommendations.<br>• Written correspondence with the patients' General Practitioner |

Karran et al. (2018), *PeerJ*, DOI 10.7717/peerj.4301

**Table 1** (*continued*)

| | GLITtER consultation | | Standard consultation | |
|---|---|---|---|---|
| WHO PROVIDED | **Clinician 1** (Physiotherapist) 14 years clinical experience 7 years' experience in SAC Post-graduate (Masters) degree Completed SAC Competencies Framework, participates in weekly professional development Completed GLITtER[a] training | **Clinician 2** (Physiotherapist) 13 years clinical experience 11 years' experience in SAC Post-graduate (Masters) degree Completed SAC Competencies Framework, participates in weekly professional development Completed GLITtER[a] training | **Clinician 3** (Physiotherapist) 17 years clinical experience 1 years' experience in SAC Post-graduate (Masters) degree Completed SAC Competencies framework, participates in weekly professional development | **Clinician 4** (Physiotherapist) 25 years clinical experience 7 years' experience in SAC Post-graduate (Masters) degree Completed SAC Competencies framework, participates in weekly professional development |
| HOW | 1:1 delivery (integrated into individual patient consultation in the SAC). 4 occasions of follow-up involving a brief SMS message. | | Individual patient consultation. | |
| WHERE AND HOW MUCH | Single session, integrated into a standard SAC consultation. Approximately 10 min's duration (in addition to standard consultation). | | Standard 'new patient' consultation in the SAC at the RAH. Approximately 30–40 min duration. | |
| TAILORING | Patient-centred standard consultation (as noted). Imaging interpretation tailored according to imaging findings. Exercise advice tailored according to the patient's age, physical condition, and practical considerations. | | Patient-centred consultation with assessment, discussion and management guided by the characteristics of the patient and their clinical presentation. | |
| MODIFICATIONS | A visual aid developed for use during the consultation was considered to be unnecessary. | | Not applicable. | |
| HOW WELL | Intervention fidelity was not assessed. (This is an important consideration for future effectiveness testing). | | Not applicable. | |

**Notes.**

GLITtER, Green Light Imaging interpretation To Enhance Recovery; LBP, low back pain; SAC, Spinal Assessment Clinic; SMS, short message service.

[a]Details of training provided to Clinicians 1 and 2 is provided in the manuscript.

## Outcomes

The primary outcome of this study was the fulfillment of specific feasibility criteria that were established *a-priori* (see Table 2) and designed to demonstrate that a subsequent effectiveness trial would be likely to be acceptable (to both patients and clinicians) and have sound methods and procedures likely to lead to successful trial completion. These criteria were considered and reported under the domains of *process*, *resource* and *scientific* considerations (*Thabane et al., 2010*) and *acceptability*. Process considerations referred to steps essential for the success of a larger study including rates of participant recruitment and retention. Resource considerations related to time and budget issues and included participant and clinician burden. Scientific considerations included estimates of treatment effect, variance of effect and data acquisition for a subsequent sample size calculation. We classified criteria that met the standards specified in Table 2 as "achieved" and criteria that failed to meet these pre-specified standards (and would need addressing or adapting before proceeding with further testing of the intervention as "review".

The secondary outcomes of this study were:

(1) Attainment of data permitting calculation of an appropriately powered sample for a subsequent RCT (i.e., the standard deviation of the NRS score for pain at 3-month follow-up).

(2) Exploratory analysis of between-group differences for changes in pain, disability, and kinaesiophobia (from baseline to 3 month follow-up).

(3) Identification of ceiling or floor effects.

(These outcomes were considered and reported under the domain of *scientific* considerations.)

## Sample size

We aimed to allocate 40 study participants to intervention or control groups using a quasi-randomised procedure. This sample size was considered adequate for a feasibility study (designed principally to assess feasibility of recruitment and procedures), and sufficient to inform a power analysis for a subsequent randomised controlled trial (*Hertzog, 2008*).

## Procedure

The full study procedure, including *preliminary* and *primary* screening procedures to determine patient eligibility for participation, is outlined in Fig. 1.

### Confirmation of participant eligibility

At completion of the SAC assessment, all clinicians completed a 5-item checklist to confirm *final* eligibility for participation in this study. The intention of this checklist was to exclude patients who required subsequent investigations, interventions or surgical intervention, or patients who had pathology requiring significant caution with activity. It also confirmed that patients had functional or exercise restrictions, such that an intervention to promote activity was warranted. Participants meeting all criteria were notified of their inclusion in the study and were informed regarding subsequent follow-up procedures.

**Table 2  Feasibility considerations, criteria and outcomes.**

| Feasibility considerations | Criteria | Outcome |
|---|---|---|
| *Process* | | |
| Recruitment rate | One participant per clinician per (weekly) clinic can be recruited (four participants recruited at each clinic) | Review |
| Consent rate | At least 70% of all eligible patients can be recruited | Achieved |
| Follow-up rate | Complete follow-up data can be obtained for at least 95% of participants | Review |
| *Resource* | | |
| Data collection: Participant burden[a] | 95% completion of baseline data (Recommended Minimum Dataset for LBP Research and TSK-11) prior to SAC appointment | Review |
| Clinician/SAC burden | consultations do not extend clinic appointments by more than 10 min (on average) | Achieved |
| *Acceptability* | | |
| Patient acceptability | >80% of responses to Questions 1–7 of the Participant Experience Questionnaire rated as "agree" or "strongly agree" | Achieved |
| Clinician acceptability & perceived benefit | SAC Clinicians (delivering the GLITtER intervention) report that they are "confident" or "very confident" when asked: ● "How confident are you that you could integrate GLITtER into standard practice on an ongoing basis?" and ● "How confident are you that integrating GLITtER would enhance SAC care?" (4-point scale) | Achieved |
| *Scientific* | | |
| Determination of sample size for appropriately powered RCT | Use the standard deviation of the NRS score for pain at 3-month follow-up to calculate sample size | Achieved |
| Exploratory analyses of effect | Calculate between-group differences for changes in pain, disability, and kinaesiophobia (from baseline to 3 month follow-up) | Achieved |
| Identify ceiling or floor effects | Presence or absence of ceiling or floor effect identified and quantified | Achieved |

**Notes.**

[a] criterion not pre-specified in protocol.

LBP, low back pain; TSK, Tampa Scale for Kinaesiophobia; SAC, Spinal Assessment Clinic; RCT, randomised controlled trial.

## Group allocation and participant blinding

Participants were allocated to the GLITtER Intervention or the control group via a quasi-randomised procedure devised to cause minimal interference to clinic processes and also allow between-group comparisons. Group allocation was determined by appointment scheduling and clinician availability, such that patients were seen in order of their arrival at the clinic by the first available clinician. Patients meeting the primary eligibility criteria were thereby pragmatically allocated to the intervention or control conditions. Clinicians 1 and 2 were advised to provide no more than one GLITtER intervention per clinic session to minimise potential time burden and disruption to clinic flow. Study participants were unaware whether they received a GLITtER consultation or a standard consultation.

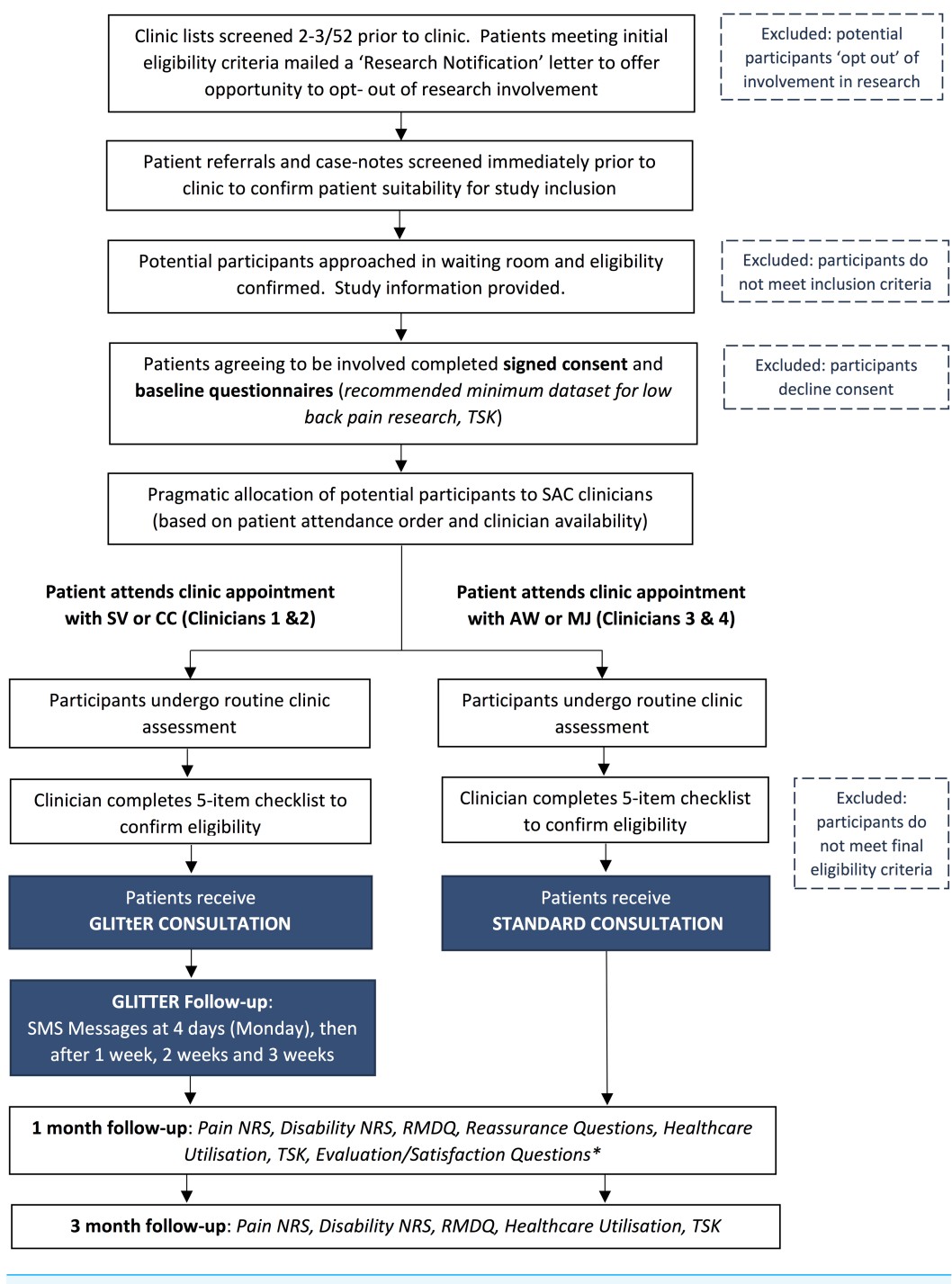

**Figure 1** **StudyFlow: Prospective, comparative, feasibility trial of GLITtER.** Abbreviations: TSK, Tampa Scale for Kinaesiophobia; SMS, short message service; NRS, numeric rating scale; RMDQ, Roland Morris Disability Questionnaire.

## Data collection

Baseline demographic and outcome data were hand-recorded by participants on purpose-designed forms. Follow-up outcome data were obtained via completion of postal questionnaires. All data were entered onto a password protected excel spreadsheet. Participants were requested to complete the Recommended Minimum Data Set for LBP Research (*Deyo et al., 2014*) and the Tampa Scale for Kinaesiophobia-11 (TSK-11) (*Tkachuk & Harris, 2012*) at study inception. At 1 month follow-up, participants were mailed numeric rating scales (NRS) for pain and disability, three questions about reassurance (Supplemental File 4), and a participant experience questionnaire (Supplemental File 5). At 3 month follow-up participants were requested to complete pain and disability NRS, the Roland Morris Disability Questionnaire (RMDQ)(*Roland & Morris, 1983*), and the TSK-11. Health care utilisation was also evaluated at 1 and 3 months (Supplemental File 5).

## Interim study evaluation

Interim evaluation of the trial was undertaken after 30% of participants had been recruited into the study. We considered problems with participant recruitment and discussed any clinician concerns or suggestions. We also reviewed participant responses for completeness and reports of intervention acceptability. Study protocol modifications were considered and implemented as considered appropriate by the research team, and all changes were recorded.

## Statistical methods

Baseline clinical and demographic characteristics of the participants were reported using descriptive statistics. Patient eligibility, recruitment and retention rates were calculated, and reasons for refused consent were recorded where they were available. Questionnaire completion rates were also calculated. The standard deviation of the primary outcome measure to be used in a future effectiveness trial (pain NRS at 3 months) was used to inform the sample size calculation for a larger RCT. We planned to conduct exploratory analysis of between-group differences in change scores for pain, disability and kinaesiophobia (from baseline to 3-month follow-up) using ANOVA, acknowledging that estimates of treatment effect should be assumed to be imprecise due to the non-randomised allocation procedure and small sample size.

# RESULTS

## Baseline characteristics of participants

Thirty one patients were enrolled in this study between the 2nd March and the 14th July, 2017, with the trial ceased due to temporary closure of the service. Detailed baseline data were collected for all participants consistent with the Recommended Minimum Dataset for LBP research (*Deyo et al., 2014*) (see Supplemental File 6). Sixty-three percent of study participants were female, and the mean age of participants was 50.1 years (standard deviation (SD) 14.0, range 20–75 years). All participants reported experiencing LBP for more than 6 months. Key baseline data are reported in Table 3.

**Table 3  Key baseline participant characteristics.**

| | Total sample ($N = 31$) | Group | |
| --- | --- | --- | --- |
| | | GLITtER consultation ($N = 20$) | Standard consultation ($N = 11$) |
| Age (mean, SD) | 50.1 (14.0) | 52.4 (11.7) | 46.1 (17.4) |
| Gender (% female) | 63% | 65% | 55% |
| Pain NRS (mean, SD): In the past 7 days, how would you rate the intensity of your pain on average? (Questions 3[a]) | 6.3 (1.8) (97% complete responses) | 6.2 (1.8) (95% complete responses) | 6.5 (2.0) (100% complete responses) |
| Disability NRS (mean, SD): In the past 7 days, how much has pain interfered with your day-to-day activities? (Questions 7[a]) | 5.8 (2.1) (100% complete responses) | 5.9 (1.8) (100% complete responses) | 5.6 (2.5) (100% complete responses) |
| Pain Catastrophising Scale (mean, SD) | 38.9 (7.3) (74% complete responses) | 40.5 (7.1) (65% complete responses) | 36.7 (7.3) (91% complete responses) |

**Notes.**

SD, standard deviation; NRS, numeric rating scale.

[a]From Recommended Minimum Dataset for Low Back Pain Research, see Supplemental Information 6.

## Interim evaluation

Interim evaluation occurred after the first 12 participants had been recruited into the trial. The main issue discussed was the failure to consistently meet the weekly target for participant recruitment. An unforeseen change to clinic process (with the result that fewer patients who were likely to be recommended for conservative management received SAC appointments) was identified as the reason for the lower number of patients meeting the final eligibility criteria than anticipated. While no significant resolution to this issue was identified, the exclusion of patients who reported a past history of spinal surgery was discussed. We recognised that many of these patients did not require further imaging or surgical opinion and had the potential to benefit from GLITtER. The primary eligibility criteria were revised to permit inclusion of participants who had had no more than 1 surgical procedure (more than 2 years prior) *and* who met all other inclusion criteria.

At this interim stage, follow-up data were available from five participants—three of whom received a GLITtER Consultation. 100% of responses on the Participant Experience Questionnaires were favourable (i.e., responded to the participant satisfaction statements with ratings of "agree" or "strongly agree"). Clinicians delivering the GLITtER intervention also reported being satisfied with study processes. They perceived the GLITtER intervention to be well accepted by patients and felt confident in their ability to deliver it time efficiently and with competence. No specific concerns were raised.

## Achievement of feasibility criteria

Achievement outcomes of the pre-specified feasibility are provided in Table 2 and detailed below.

### *Process considerations*

A total of 31 participants were recruited from 15 clinic sessions. A CONSORT flow diagram detailing study recruitment is provided in Fig. 2. Of the 101 patients who met the primary eligibility criteria, 24 patients declined involvement. This resulted in a consent rate of 75%,

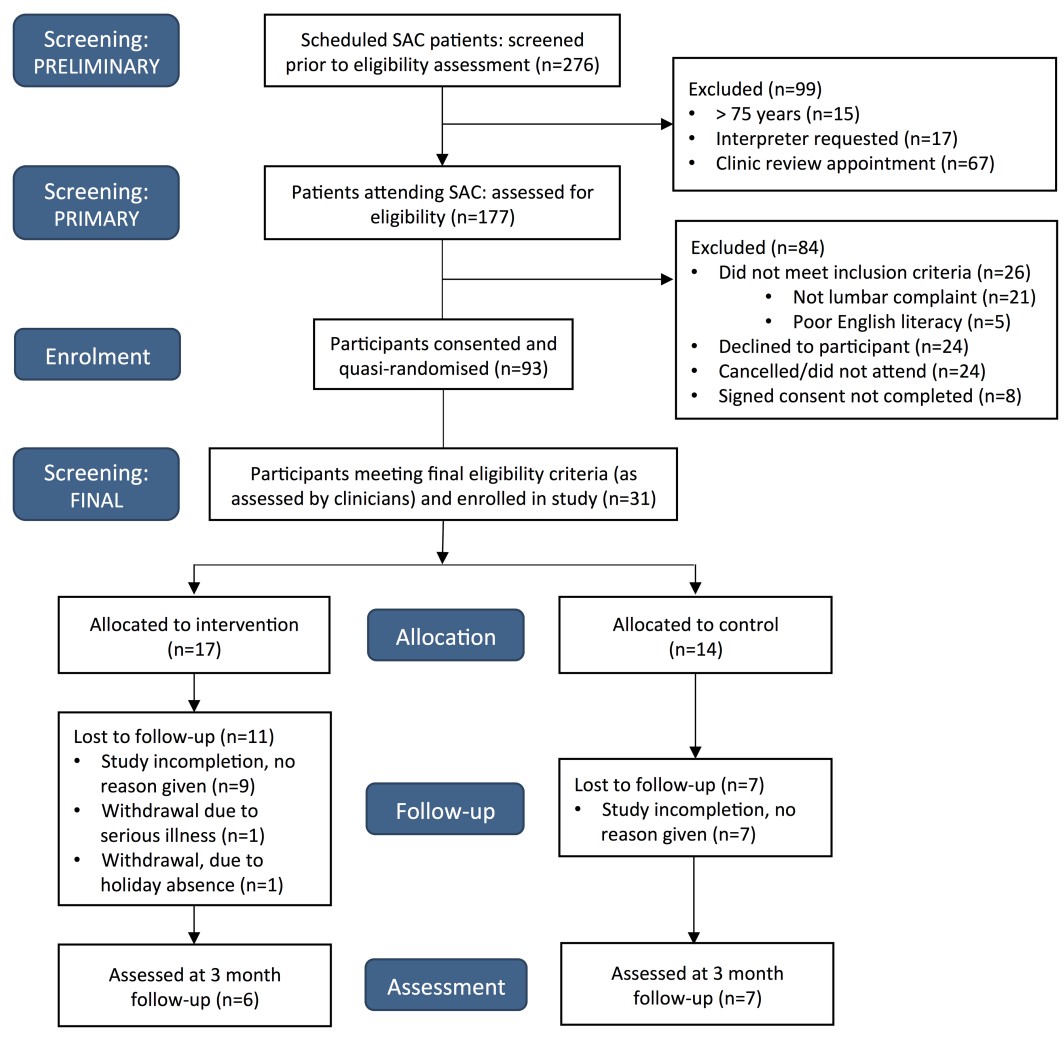

**Figure 2   CONSORT flow diagram.**

which was considered acceptable according to our pre-specified criteria. In addition to this however, 8% did not complete signed consent forms due to unspecified reasons. An average of two participants were recruited each clinic session, which was much less than our target recruitment rate of four participants per clinic session. Further consideration and revision of recruitment feasibility is required. Acceptable participant follow-up response rates were also not achieved, with 1 and 3-month outcome data available for 36% and 42% of participants respectively.

### Resource considerations

Participants completed 95% of all data items at baseline. 96% of items on the Recommended Minimum Dataset were completed and 92% of items on the TSK-11 were completed. Clinician burden was evaluated by the duration of the GLITtER consultations in comparison with standard consultations. 54 SAC consultations (11 GLITtER consultations and 43

**Table 4** Baseline, 3 month follow-up and change scores for pain, disability and kinaesiophobia.

| | Baseline scores | | 3 month follow-up | |
|---|---|---|---|---|
| | GLITtER mean (SD), N | Standard mean (SD), N | GLITtER mean (SD), N | Standard mean (SD), N |
| Pain (NRS) | 6.2 (1.8) N = 19 | 6.5 (2.0) N = 7 | 6.5 (2.0) N = 11 | 5.3 (2.1) N = 6 |
| Disability (NRS) | 5.9 (1.8) N = 20 | 4.2 (2.8) N = 7 | 5.6 (2.5) N = 11 | 4.2 (2.9) N = 6 |
| Kinaesio phobia (TSK 11) | 40.6 (7.1) N = 13 | 38.9 (7.2) N = 5 | 36.7 (7.3) N = 10 | 30.0 (7.8) N = 5 |

**Notes.**
SD, standard deviation; NRS, numeric rating scale; TSK, Tampa Scale for Kinaesiophobia.

Standard Consultations) were timed by clinicians 1 and 2. The mean duration of the GLITtER consultations was 41 min (SD = 7, minimum 30, maximum 50). The mean duration of standard consultations was 40 min (SD = 9 min, minimum 25, maximum 70). Three of the standard consultations exceeded 50 min. If these consultations were excluded, the mean duration of the Standard Consultations was reduced to 39 min (SD = 7).

### Management considerations

100% of responses on the participant experience questionnaires (for participants who received a GLITtER consultation) were favourable (i.e., participants responded to the satisfaction statements with ratings of "agree" or "strongly agree"). Clinicians 1 and 2 both reported that they were "confident" that they could integrate GLITtER into standard practice on an ongoing basis (on a 4-point Likert scale ranging from "not at all confident" to "very confident"). One clinician was "very confident" and the other was "confident" that integrating GLITtER into routine consultations would enhance SAC care.

### Scientific considerations

Based on a linear mixed effects model with two time points (baseline and 3 months), power = 80%, Type 1 error = 5%, an expected correlation between baseline and 3 months measurements of 0.5, a MCID of 1, and a SD of 2.1, then 53 patients are required per group. Allowing for 15% loss to follow-up, a sample size of 63 participants per group is required for an appropriately powered RCT.

Mean pain, disability, and kinaesiophobia scores at baseline and 3 month follow-up are provided in Table 4. We did not calculate change scores or conduct exploratory analysis of between-group differences (as specified a priori) due to the small sample sizes achieved in each group. No ceiling or floor effects were observed.

## DISCUSSION

This investigation has highlighted some important operational challenges impacting the feasibility of successfully conducting a future effectiveness trial of GLITtER (Green Light Imaging Interpretation to Enhance Recovery). While pre-specified targets for consent rates, resource burden and acceptability of the GLITtER intervention were met, participant recruitment was slower than anticipated and an acceptable follow-up rate was not achieved.
Recruitment of participants into the study occurred at half of the anticipated rate, due largely to fewer patients meeting the final eligibility criteria than expected. An identified contributor to this was the change to referral management processes such that patients who were unlikely to be candidates for intervention were no longer offered SAC appointments. (Instead, these patients and their General Practitioners were informed via a letter that community-based management was most appropriate.) Although the number of patients attending the SAC who were eligible for this study was low, this may not indicate that the number of individuals suitable for the GLITtER intervention is also low. Instead, it may indicate that there is need to further consider how best to access the patients for whom GLITtER may offer potential benefit. Alternatively, plans to further investigate the GLITtER intervention in the spinal outpatient setting must carefully consider patient triage procedures, allow for slower recruitment, and investigate the potential for recruitment at multiple sites. The wording on the information sheet should also be reconsidered to avoid the risk that some patients may have declined participation out of concern that they would be given reassuring information during their consultation, rather than the information that they were seeking.

We achieved a 3-month participant follow-up rate of only 42%, which failed to meet our feasibility criterion. In hindsight, our objective to achieve complete follow-up data from 95% response rate was high—a drop-out rate of more than 20% is generally considered to compromise the validity of a clinical trial unacceptably (*Schulz & Grimes, 2002*). Possible reasons for the high drop-out rate may have included participant improvement (such that LBP was no longer a significant concern), or patient perceptions of a lack of relevance of the intervention or the follow-up requests. The protocol for collecting follow-up data via postal questionnaires and SMS reminders was based on the procedure applied in our previous work, which achieved a 4 month follow-up rate of 89% (*Karran et al., 2017b*). There were two main differences in the follow-up conditions between these studies. Firstly, our prior study involved participants while they were still waiting for their clinic appointments to be scheduled which may have facilitated engagement, whereas follow-up for the current trial occurred after patients had received their clinic consultation. Secondly, for our initial study, we sent small packets of confectionery with requests for completion and return of questionnaires, whereas in the current trial we did not. Future investigations in this setting should further consider developing an enhanced protocol for achieving follow-up targets and providing participant incentives (*Brueton et al., 2013*). It may also be important to consider revision of the inclusion criteria to ensure that the patient's problem is of sufficient severity to warrant engagement with the study. Unfortunately, the poor response rate attained in this study impacted the achievement of our secondary objectives. Collected data were used to cautiously inform the determination of sample size for a larger RCT, but was considered insufficient to undertake exploratory analysis of effect.

The GLITtER intervention was acceptable, of perceived benefit to both patients and clinicians, and was able to be integrated into a standard consultation without significant time burden for clinicians. The 95% data completion rate on the recommended minimal dataset for LBP research (*Deyo et al., 2014*) suggests that despite its length, it is not overly onerous for participants. The consent rate of 75% exceeded our target rate, however it must

be acknowledged that a further 8% of patients who met the primary eligibility criteria did not either confirm or decline consent. Whilst a number of these exclusions were likely to be related to procedural or timing issues it must be also considered that some of these patients may have opted out of the study (without this action being recorded)—with impact on the overall consent rate.

This study employed rigorous scientific methods throughout. The protocol was developed in accordance with the SPIRIT checklist and pre-registered on Open Science Framework. The trial was registered on the ANZCTR and has been reported according to the CONSORT statement. Description of the GLITtER intervention followed the TIDieR recommendations for reporting of clinical interventions and the NIHs task force recommended minimum data set for LBP clinical trials was implemented. The specification of feasibility criteria prior to conducting this investigation is a further strength of this study's design.

Several weaknesses have also been recognised. There are no standardised criteria for evaluating feasibility, or how achievement of the established criteria should be measured. We used an evidence-informed but ultimately investigator-led approach to identify and quantify our feasibility criteria, which may not have been optimal. Factors such as participant adherence to management recommendations and investigator burden could also have been considered. In addition, we consider that the ability to engage clinicians and administrative personnel is imperative to the success of a clinical trial but we did not formally assess engagement. Our sample size and our follow-up rate were smaller than we expected, which resulted in less data to evaluate our feasibility criteria than we specified as a requirement a priori. The study sample may therefore not be representative of the larger patient population, raising the risk that estimates or interpretations that arise lack precision or are potentially misleading. Last, staff training was not formalised and treatment fidelity was not assessed. While we did calculate an estimated sample size required for a RCT to definitively examine the effect of the GLITtER intervention, this result should be interpreted with caution. Encouragingly, the baseline value for the pain NRS score and the standard deviation of this measure at 3-month follow-up, were very similar to the data recorded in another LBP feasibility trial on which a sample size estimate was also based (*Ellard et al., 2017*). Based on the assumption that two participants from each SAC would meet final eligibility criteria for inclusion in the study and be randomised, and that clinics operate on 46 weeks of the year—it appears realistic that 92 participants could be recruited annually. Timely achievement of a recruitment target of 126 is likely to be possible, and potential also exists to consider enrolment of participants at two or more sites.

This study successfully tested for feasibility of a highly pragmatic approach to addressing the clinical challenge of providing time-efficient, low cost, guideline-informed management of patients with LBP attending secondary care. Based on analysis of our feasibility criteria, a full-scale RCT evaluating the effectiveness of the GLITtER intervention is not recommended without considering strategies that are capable of resolving the recruitment and follow-up issues that we have identified. The shift in clinical pathways raises the possibility that testing of GLITtER in primary care may be indicated. These issues are likely to affect other trials in this population and as such, we humbly suggest that the lessons we have learnt be considered recommendations for the field.

## CONCLUSIONS

Failure to achieve pre-specified recruitment and follow-up rates was disappointing, however these were important outcomes of a feasibility study. It is realistic to consider that these challenges can be overcome through careful strategy, ample funding and continued partnership with health care providers.

## ACKNOWLEDGEMENTS

The authors gratefully acknowledge the cooperation and contributions of staff from the Royal Adelaide Hospital Spinal Unit, Spinal Assessment Clinic and Physiotherapy Departments who made this study possible. We also express our thanks to the patients involved.

### Funding

Emma L. Karran was supported by the Royal Adelaide Hospital Research Foundation Clinical Research Grant (2015) and the Royal Adelaide Hospital Research Foundation Dawes Scholarship (2016–2018). G. Lorimer Moseley is supported by the National Health and Medical Research Council (NHMRC), Australia (ID: 106279). The funders had no role in study design, data collection and analysis, decision to publish, or preparation of the manuscript.

### Grant Disclosures

The following grant information was disclosed by the authors:
The Royal Adelaide Hospital Research Foundation Clinical Research Grant (2015).
The Royal Adelaide Hospital Research Foundation Dawes Scholarship (2016–2018).
The National Health and Medical Research Council (NHMRC), Australia: 106279.

### Competing Interests

G. Lorimer Moseley has received support from Pfizer Australia, Workers' Compensation Boards in Australia, North American and Europe, NOIgroup Australasia, Kaiser Permanente California, Results Physiotherapy, Agile Physiotherapy, the International Olympic Committee and the Port Adelaide Football Club, and receives royalties from the following books: Explain Pain; Explain Pain Handbook: Protectometer; Explain Pain Supercharged; Painful Yarns—Metaphors and Stories to Help Understand the Biology of Pain; the Graded Motor Imagery Handbook. All other authors declare that they have no competing interests.

### Author Contributions

- Emma L. Karran conceived and designed the experiments, analyzed the data, wrote the paper, prepared figures and/or tables, provided training for clinical staff.
- Susan L. Hillier conceived and designed the experiments, analyzed the data, reviewed drafts of the paper.

- Yun-Hom Yau conceived and designed the experiments, analyzed the data, reviewed drafts of the paper, provided training for clinical staff.
- James H. McAuley and G. Lorimer Moseley conceived and designed the experiments, reviewed drafts of the paper.

## Clinical Trial Ethics

The following information was supplied relating to ethical approvals (i.e., approving body and any reference numbers):

The Research Ethics Committees at the Royal Adelaide Hospital (protocol no. 150308) and the University of South Australia (protocol no. 0000034887) provided approval for this research.

## Clinical Trial Registration

The following information was supplied regarding Clinical Trial registration:

Australian and New Zealand Clinical Trials Registry: ACTRN12617000317392.

## Supplemental Information

Supplemental information for this article can be found online at http://dx.doi.org/10.7717/peerj.4301#supplemental-information.

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
