# Peer review of "A quasi-randomised, controlled, feasibility trial of GLITtER (Green Light Imaging Interpretation to Enhance Recovery)—a psychoeducational intervention for adults with low back pain attending secondary care"

_PeerJ, doi:10.7717/peerj.4301_

## Round 0.1 · original submission · Major Revisions

1. The dates of the study are not given in the text.

2. Sample size calculation should be given with more particulars: power, alpha, expected parameters... Furthermore, you have indicated it is necessary to have 40 patients and your final sample size is 31 for base-line situation and 13 for the final of the follow-up.

Please address all the comments of the reviewers.

·

Basic reporting

The writing of the manuscript is clear and well written. You have used appropriate references to set the scene for the study and also to support your findings. The article is well structured and the tables are very descriptive. Although I wonder if there is perhaps a too much detail included in Table 1? I think the "Who provided" part can be summarised and included in the methods section rather than in Table 1.
The results clearly address the aims of the study and you have considered reasons for the challenges encountered.

Experimental design

The research is original, important and the question itself is well defined. Reassurance of a favourable outcome is unfortunately not commonly provided to people with LBP, so I commend the authors for developing a strategy to help address this healthcare issue. I think there are some areas of the methods section which are thin and can be improved. Detailed comments in general comments for the authors section. I think the adaptive design was a great idea.

Validity of the findings

I think the findings are encouraging and will certainly help to inform a future RCT in the area. The conclusions are well supported by the results, and link back to the original aims.

Additional comments

This is a well designed feasibility study exploring the acceptability of the GLITtER intervention for people with low back pain attending an outpatient clinic. It’s a shame recruitment rate was slower than you had hoped however I hope that the planned adjustments will be useful for the future RCT. I have a few suggestions for the authors to consider.
1. I think the paper could be improved by clearly specifying in the methods section what was considered acceptable or not acceptable for each domain being evaluated. For example in the discussion section you mention “We achieved a 3-month participant follow-up rate of only 42%, which failed to meet our feasibility criterion.” However what was acceptable for this criterion? I would suggest that some more description is needed regarding how each criterion was judged as being “met” or requiring “review”.
2. I think there could be some more description in the methods section on the actual intervention - GLITter, what is meant by “standard care”, and how the proposed intervention differs to standard care in the context of this study. I suggest that the key information from Table 1 be summarised and included in the main text also.
3. With regards to the outcomes – what is the rationale behind choosing the outcomes? I suggest including just s brief sentence highlighting the importance of the outcomes selected – this will be particularly useful for readers with a non-physiotherapy or medical background.
4. You specify that the target population is patients with low back pain however could you please clarify whether this includes both acute and chronic LBP? Evaluation of patients may differ depending on duration of pain and number and nature of prior healthcare visits – was this sort of information considered in the feasibility study?
5. I like the name of the intervention – GLITtER. Some times however I felt like I needed to be reminded what it stood for – perhaps in sections like the first paragraph of the discussion, the actual name of the intervention could be used to reinforce what it was all about.
I have a few more minor comments:
Results: Where it says an average of 2.1 participants on page 9 line 281 – I recommend just saying 2 participants.
Discussion: Please review the sentence on page 11 line 331.

·

Basic reporting

Language OK
Literature/references OK
Article structure/ tables / figures OK
Relevant results: Discussion and conclusions should be improved, please see below

Experimental design

Relevant for the aims and scope of the journal OK
Research questions well defined OK
Rigorous investigation and technical standards OK
Methods not sufficiently described, please see below
Design may not be optimal, please see below

Validity of the findings

Impact and novelty OK
Data robust and statistically sound OK
but impaired by low follow-up rate

Additional comments

The Introduction is well written and points to an important issue to be addressed.

Methods:
To the present reviewer it is not clear how this lot of information for the patient in the GLITtER intervention arm can be delivered within 40 minutes.
Furthermore, I wonder whether it is possible to deliver a control intervention not including the new knowledge about the pain system and inconsistent associations between findings on spinal imaging and back pain. Most therapists know about this, and I find it difficult to imagine how a therapist can deliver a standard control intervention without involving this knowledge.

Discussion:
The Discussion should be shortened and in stead issues presented below may be considered.
The two main problems were the slow recruitment and the low follow-up rate.

Slow recruitment:
According to the present reviewer, the information sheet for patients was not appropriate.
We know reassurance is important as well as worrying and health anxiety1. However, a patient would not like to be met with "the need for reassurance" as an objective. The patients referred for non-urgent secondary health care usually has a long-lasting pain problem interfering with their life. They want to know: What is the cause of pain and what can be done (and perhaps also: is it dangerous and what about the prognosis?).

Proposal for another wording of the information sheet:
It is well recognized worldwide that relevant information for LBP patients is essential for achieving good results. New knowledge about pain in general and the relation between structural changes on spinal imaging and pain is now available. This research is interested in testing whether a more detailed information package for patients may affect recovery. The participants will receive one of two information packages and follow-up will be arranged. The standard treatment will not differ between groups.


Low follow-up rate:
This is possibly the most difficult problem to solve, and a solution is essential for a large scale study to succeed.
Possible explanations:
The back problem has disappeared or has changed to a minor problem. Usually, it is the younger and less sick patients that do not respond to follow-up.
Another explanation: Patients do not think that the health care delivered was relevant for their problem.

Suggestions:
Securing that the problem is sufficiently severe by demanding some kind of sick-listing as supplemental inclusion criteria. Sick-listing and loss of contact to the labor market are the most important issues in non-specific LBP.
Arrange a control visit after three months indicating that the problem is taken seriously. If the patient does not show up, he or she may be contacted.

Consideration on ethics:
Is it ethically sound not to inform a patient about the present knowledge about the pain system and the poor association between findings on spinal images and pain as well as prognosis?
Suggestion for design solving this problem: Arranging a follow-up visit after 3 (or 6) months. The patients not receiving the new package in the first place will have the information package in the second place.
Considerations on design: The low rate of follow-up makes it difficult to conclude anything regarding differences in outcome between intervention groups. However, the present reviewer think that there would be a high risk for a randomized trial to show no differences in outcome between the two interventions, especially because of the considerations mentioned in the Methods section above.
Is the proposed randomized design the optimal design? Another kind of randomized design would be to deliver a standard intervention, and then contacting the patients for inclusion in a randomized study delivering a supplemental intervention by another organization, a study design used elsewhere2. A population intervention comprising two lectures on the workplace was effective in reducing sick-listing due to LBP the following year3. Hence, that study showed how important relevant knowledge about back pain is for appropriate back pain behavior. Would a population strategy be more relevant?
Minor issues:
In Figure 2 it looked like every patient receiving two clinical interventions, not one. Why?
What were the "pragmatic reasons" for stopping inclusion after 4 months?
Should available MRI also be one of the inclusion criteria?


References
1. Jensen OK, Nielsen CV and Stengaard-Pedersen K. One-year prognosis in sick-listed low back pain patients with and without radiculopathy. Prognostic factors influencing pain and disability. Spine J 2010; 10: 659-675.
2. Lambeek LC, van Mechelen W, Knol DL, et al. Randomised controlled trial of integrated care to reduce disability from chronic low back pain in working and private life. BMJ 2010; 340: c1035.
3. Frederiksen P, Indahl A, Andersen LL, et al. Can group-based reassuring information alter low back pain behavior? A cluster-randomized controlled trial. PLoS One 2017; 12: e0172003.

---

## Round 0.2 · accepted · Accept

Dear authors,

I am happy to inform you that your paper has sufficient standards to be published in PeerJ in its current form.

Congratulations!

Warm regards,
Dr Palazón-Bru (academic editor for PeerJ)

·

Basic reporting

All domains acceptable

Experimental design

All domains acceptable

Validity of the findings

All domains acceptable

Additional comments

You have adequately addressed my comments, thank you.

·

Basic reporting

OK

Experimental design

OK

Validity of the findings

OK

Additional comments

Although major revisions have not been performed in the manuscript, I am satisfied with the adjustments in the manuscript, and I am glad that the authors seem to have taken into account my critical points and will include these considerations in the planning of a randomized controlled trial. I am not able to see my references at the end of their answer, but apparently the authors have seen the references.
It is very important that the design of a future randomized trial is optimal, so that the study will show a realistic picture of the importance of the information package for LBP patients. To the present reviewer there is a high risk of finding no difference in outcome between the two interventions as they are described at present. A negative result will have great negative impact of the whole discussion regarding LBP patients´ need for relevant information.